# The significance of anxiety symptoms in predicting psychosocial functioning across borderline personality traits

Jacqueline Howard[1], Robinson De Jesu´s-Romero[1], Allison Peipert[1], Tennisha Riley[2], Lauren A. Rutter[1], Lorenzo Lorenzo-Luaces[1] *

**1** Department of Psychological and Brain Sciences (PBS), Indiana University-Bloomington, Bloomington, Indiana, United States of America, **2** School of Education, Indiana University-Bloomington, Bloomington, Indiana, United States of America

* lolorenz@iu.edu

**Data Availability Statement:** The data are available at the Open Science Framework: https://osf.io/7hcpq/.

## Abstract

Emotion regulation is a central task of daily life. Difficulty regulating emotions is a core feature of borderline personality disorder (BPD), one of the most common and impairing personality disorder diagnoses. While anger and symptoms of depression are instantiated in the criteria for BPD, anxiety is not, despite being among the most common psychiatric symptoms. In a sample of online respondents (N = 471), we explored the interactions between anxiety symptoms and BPD traits in predicting well-being (WHO-5) as well as poorer work and social adjustment (WSAS), while controlling for anger and depression. We hypothesized that anxiety would lead to more impairment (i.e., lower well-being and poorer work and more difficulties with work and social adjustment) as BPD traits increased. BPD traits and symptoms of anxiety both contributed to overall lower levels well-being and higher levels of psychosocial dysfunction. However, contrary to our expectations, at higher (vs. lower) levels of BPD traits, symptoms of anxiety were less conducive to lower well-being on the WHO-5. For the WSAS, there was no consistent evidence for an interaction between BPD traits and anxiety in predicting functioning. By and large, our results do not support the idea that anxiety contributes to more impairment at higher levels of BPD traits.

## Introduction

Emotion regulation is a core aspect of healthy mental functioning. The concept has been more concretely defined as "things we do to influence which emotions we have, when we have them, and how we experience and express them" [1]. Deficits in emotion regulation contribute to the development and maintenance of many psychological problems [2–4] and are considered a hallmark feature of borderline personality disorder (BPD) [5, 6]. BPD is characterized by instability in moods, behavioral dysregulation (e.g., impulsivity, self-injurious behaviors), and interpersonal difficulties [7]. At least 6% of the population may meet diagnostic criteria for BPD [8]. Although the current criteria identify the presence (vs. absence) of BPD as if it were a category, the disorder and its symptoms are better thought of on a continuum [9, 10]. BPD traits, even a single BPD trait, have a significant impact on functioning [11, 12]. Even though

**Funding:** Ms. Peipert and Mr. De Jesús-Romero were funded by NIMH grant T32 MH103213. The funders had no role in study design, data collection and analysis, decision to publish, or preparation of the manuscript.

**Competing interests:** The authors have declared that no competing interests exist.

the long-term course of BPD can be variable, including modest rates of symptom remission and functional recovery over long-term follow-ups [13, 14], in psychiatric settings BPD is one of the most "fatal" disorders with high rates of suicide, self-injurious behaviours, and hospitalizations [15, 16].

The biosocial model of BPD proposed by Linehan [5, 6] posits that BPD traits emerge from a generalized vulnerability to intense states of negative affects coupled with a dearth of skills to regulate this affect, usually as the product of an invalidating upbringing [17]. The Diagnostic and Statistical Manual of Mental Disorders (DSM) criteria for BPD directly address difficulty generally regulating emotions (i.e., affective instability). Above and beyond affective instability, the criteria also reference specific problems controlling anger (i.e., inappropriate, intense anger, or difficulty controlling anger) as well as symptoms that could be thought of as related to depression (e.g., emptiness, suicidal ideation) [7]. Interestingly, the BPD criteria do not explicitly address symptoms of anxiety. Anxiety not being a part of the diagnostic criteria of BPD may simply be an artifact of DSM-5 categorical classification. However, it is curious that the criteria do not include a reference to feelings of anxiety, despite the finding that anxiety disorders are among the most common mental health disorder diagnoses in the general population [18, 19] and often occur in conjunction with personality disorder (PD) symptoms [8, 20, 21]. As much as one fifth of the population in the United States may recall meeting the criteria for an anxiety disorder at some point in their life [18, 22].

These rates may be higher in prospective studies with multiple waves, given evidence that individuals may fail to recall periods of anxiety and depression [23, 24].

Existing research on BPD supports the idea that anxiety disorders meaningfully contribute to the clinical profile in BPD. For example, in one study of 409 patients, those with BPD were more likely to report Axis I disorders, and especially multiple comorbid disorders like depressive, bipolar, anxiety, substance use, eating, and somatic symptom disorders, than patients with no BPD diagnosis [20]. Other studies have reported higher rates of anxiety in BPD patients than in those with no BPD [21]. In one study of inpatients with BPD, for example, anxiety disorders were as common among patients with BPD as depressive and bipolar disorders, but anxiety was more strongly associated with BPD than other disorders [25]. A similar finding was reported by Gratz and colleagues [26] who reported that anxiety sensitivity, and the fear/distress associated with thoughts, feelings, and behaviors related to anxiety, distinguished BPD from non-BPD patients.

BPD traits are associated with lower work and social functioning [11, 12], and there is evidence to suggest that BPD is more impairing than other PDs. For example, Zanarini [14] reported that rates of long-term symptom remission in BPD were comparably high in BPD vs. other PDs, but that BPD had lower rates of functional recovery. Based on our clinical observations as well as how common anxiety symptoms are, we hypothesized that symptoms of anxiety would be greater contributors to dysfunction among individuals the more BPD traits they endorsed (i.e., anxiety would be a greater predictor of dysfunction among individuals high vs. low on BPD traits). The role of anxiety symptoms in the context of high BPD traits is important to understand in part because patients with BPD are often excluded from randomized controlled treatment trials [27, 28]. To our knowledge, the interaction of BPD traits and anxiety in predicting general functioning has not been studied.

## Materials and methods

### Participants

Participants were recruited from Qualtrics panels. Individuals recruited by Qualtrics saw an advertisement for a study on "emotions in everyday life." Participants were given the

opportunity to read a Study Information Sheet, delineating details of the study, risks and benefits, confidentiality, and payment. Participants indicated informed consent by affirming "Click here if you understand that by moving onto the next page, I am consenting to participate in this study." Participants were deemed eligible to participate if they were a) between the ages of 18–81 years old, b) committed to giving correct answers to the best of their ability, c) passed a reCAPTCHA, and d) spent at least 5 minutes on the survey. We aimed to recruit a sample that contained at least 40% male or female participants (i.e., either 60/40 male/female, or 60/40 female/male). Prior work supports Qualtrics as a source of nationally representative data as well as a source of reliable responding [29]. The study was approved by Indiana University's institutional review board.

## Measures

**Dependent variables.** We obtained two measures of functioning, one more focused on the perception of overall positive enjoyment and another focused on negative aspects of functioning (i.e., problems at work and in social relations).

*World Health Organization—5 Well-being Index (WHO-5)* [30]. The WHO-5 is a 5-item self-report measure that assesses perception of quality of life over the past two weeks. Example items include "I have felt cheerful and in good spirits" and "My daily life has been filled with things that interest me." The items are rated on a 6-point Likert scale ranging from 0 (At no time) to 5 (All of the time), producing raw scores that range on a scale from 0–25 and are scored on a 0–100 scale by multiplying by four. Prior work supports the reliability and validity of WHO-5 ratings [31]. In the current sample, the scale obtained a Cronbach's alpha of 0.92.

*Work and Social Adjustment Scale (WSAS)* [32]. The WSAS is a 5-item self-report measure that assesses functional impairment in work, relationships, household and leisure activities as a result of a specific challenge (in this case "difficulty managing my emotions"). Example items include "Because of my difficulty managing my emotions my ability to work is impaired" and "Because of my difficulty managing my emotions my ability to form and maintain close relationships with others, including those I live with, is impaired." Items are rated on an 9-point Likert scale ranging from 0 (Not at all) to 8 (Very severely), producing scores that range on a scale of 0–40. Prior work supports the reliability and validity of WSAS ratings across various patient populations [33, 34]. In the current sample, the scale obtained a Cronbach's alpha of 0.95.

**Symptoms.** *BPD symptoms*: *The McLean Screening Instrument for BPD (MSI-BPD)* [35]. The MSI-BPD is a 10-item self-report measure assessing symptoms of BPD over the past several years. The MSI-BPD includes all the symptoms of BPD but distinguishes between paranoia and dissociation, thus containing ten items instead of nine. Example items include "Have you been extremely moody?" and "Have you chronically felt empty?" Each item is either endorsed or denied (i.e., 1 or 0), producing a range of scores from 0–10. Prior work supports the reliability and validity of MSI-BPD ratings [36]. In the current sample, the scale obtained a Cronbach's alpha of 0.89.

*Depression, anxiety, and anger.* Symptoms of depression, anxiety, and anger were evaluated with the PROMIS (Patient-Reported Outcomes Measurement Information System) emotional distress depression, anxiety, and anger short-forms from the DSM cross-cutting measures. Prior work supports the reliability and validity of the PROMIS Emotional Distress measures [37, 38]. All raw scores were converted into t-scores according to the published norms.

*PROMIS emotional distress, depression, short form* [7, 39]. The PROMIS depression measure is an 8-item self-report measure designed to assess depression severity over the past week. Examples items include: "I felt depressed" and "I felt hopeless." Items are rated on a 5-point

Likert scale ranging from 1 (Never) to 5 (Always), producing scores that range from 0–40. In the current sample, the scale obtained a Cronbach's alpha of 0.96.

*PROMIS emotional distress, anxiety, short form* [7, 39]. The PROMIS anxiety measure is a 7-item self-report measure that examines the frequency and severity of anxiety symptoms over the past week. Example items include: "I felt worried" and "I found it hard to focus on anything other than my anxiety." Items are rated on a 5-point Likert scale ranging from 1 (Never) to 5 (Always), producing scores ranging on a scale from 0–35. In the current sample, the scale obtained a Cronbach's alpha of 0.95.

*PROMIS emotional distress, anger, short form* [7, 39]. The PROMIS anger measure is a 5-item self-report measure that captures the frequency and severity of anger over the past week. Example items include: "I felt angry" and "I felt like I was ready to explode." Items are rated on a 5-point Likert scale ranging from 1 (Never) to 5 (Always), producing scores that range on a scale of 0–25. In the current sample, the scale obtained a Cronbach's alpha of 0.94.

*Perception of negative affect*. Perceptions of negative affect were evaluated through a single question asking participants which emotion (anxiety, depression, or anger) was most bothersome, or if every emotion felt out of control.

## Demographics

Information about participant age, gender (i.e., male, female, trans, or non-binary), education (i.e., <HS, HS, some college, Associate's, Bachelor's, Master's, Doctoral/Professional degree), sexual orientation (i.e., heterosexual vs. LGBTQ), relationship status (i.e., single, dating, married, divorced, or widowed), race (i.e., non-Hispanic White vs. racial-ethnic minority), ethnicity (i.e., Hispanic vs non-Hispanic), and SES (i.e., household income range from <$15,000-$200,000+) were collected.

## Missing data

Rates of missing data were low. Across variables, missing data ranged from a low of 0% (e.g., age, gender) to a high of 10.19% (i.e., self-reported difficulty with emotions) with an average rate of missing data across variables of 4% (SD = 3.68%, Median = 2.97%). Using listwise deletion would have eliminated less than 13% of the sample having missing data on at least one variable (n = 61). Nevertheless, to avoid biases associated with "completers only" analyses, we imputed missing data using the imputation by random forests with the "missForest" package in R [40].

## Data analysis

First, we conducted descriptive analyses presenting average levels of borderline symptoms (MSI), depression, anxiety, and anger (on the PROMIS measures), well-being (WHO-5), and work and social functioning (WSAS). In this sample, depression, anxiety, and anger were very highly correlated (See Table 1), even without correcting for attenuation ($r$ s > 0.78). To be able to assess the relative effects of depression, anxiety, and anger on outcomes, we conducted an exploratory factor analysis (EFA) of the 20 items from the PROMIS Emotional Distress scales, conducting a parallel analysis to determine the number of factors [41]. The results (see S1 Table) showed a general separation between symptoms of depression, anxiety, and anger. For all inferential analyses, unless otherwise stated, we use the factor scores from this EFA which are standardized like Z-scores (i.e., with a mean of zero) and typical range from -3 to +3 (derived from Varimax rotation of this EFA) to represent symptoms of anxiety, depression, and anger [42]. Next, we conducted two regressions. First, we regressed well-being on the WHO-5 on the main effects of the depression, anxiety, anger, and BPD traits. Next, we

**Table 1. Correlations of borderline symptoms, functioning, and internalizing disorder symptoms (n = 471).**

|  | MSI-BPD | WHO-5 | WSAS | Depression | Anxiety |
|---|---|---|---|---|---|
| WHO-5 | -0.28 |  |  |  |  |
| WSAS | 0.51 | -0.03 n.s. |  |  |  |
| Depression | 0.62 | -0.37 | 0.52 |  |  |
| Anxiety | 0.53 | -0.32 | 0.47 | 0.82 |  |
| Anger | 0.60 | -0.31 | 0.50 | 0.78 | 0.80 |

Note. All correlations significant at $p < 0.05$, unless otherwise stated. n.s. = not statistically significant at $p < 0.05$, WHO-5 = WHO-5 Well-Being Index, WSAS = Work and Social Adjustment Scales, Depression = MSI-BPD = self-reported McLean Screening Instrument for Borderline Personality Disorder, Depression = Patient Reported Outcomes Measurement Information System (PROMIS), Emotional Distress—Depression, Anxiety = PROMIS—Emotional Distress—Anxiety, Anger = PROMIS—Emotional Distress—Anger

regressed work and social functioning on the WSAS on the main effects of the depression, anxiety, anger, and BPD traits.

To test our main question, whether symptoms of anxiety were greater predictors of functioning among individuals the more BPD traits they endorsed, we regressed well-being on the WHO-5 on the main effects of the depression, anxiety, anger, and BPD traits, as well as on the interactions of BPD traits with depression, anxiety, anger. We ran the same model with the WSAS as the dependent variables. In these models, the anxiety*BPD term indicates whether BPD moderates the association between anxiety and well-being/functioning [43]. If anxiety interacts with BPD traits at $p < 0.05$, it would indicate that anxiety is more (or less) predictive of well-being and functioning in people higher than lower on BPD traits. We hypothesize that anxiety will be more impairing among people with BPD. We are primarily interested in the relationship between BPD traits and anxiety in predicting well-being or functioning. We also tested the interactions between depression and anger as these are potential confounds, but we do not correct for multiple comparisons because we are not making inferences about depression and anger. If the interactions between anxiety and BPD traits in predicting well-being or work and social adjustment were statistically significant, we used the Johnson-Neyman technique to probe specific levels at which to explore the association between anxiety and BPD features [44]. At the request of a reviewer, we controlled for gender identity as a binary (men vs. women) in all analyses. Three participants identified as transgender and were grouped with the gender they identified. Following recommendations, binary variables were coded as (+/- 0.5) and ordinal variables (i.e., the MSI-BPD) were centered at their median [45].

## Results

A total of 471 responders provided valid answers to our survey (i.e., excluding bots, those who did not pass the CAPTCHA, speedy responders, and those who did not commit to answering the survey truthfully). The average age was 46.76 (SD = 16.34). Most identified as either male (52.4%, n = 247) or female (46.9%, n = 221). Most were Non-Hispanic White (74.31%, n = 350). Most had completed at least some college or an associate degree (60.9%, n = 287), with very low representation from individuals who had less than a high school education (3.8%, n = 18) or with a doctoral/professional degree (3.6%, n = 17). The median household income category was $50, 000 - $74, 999 (67.30%, n = 317), though only 20.38% of people belonged to that income group (n = 96) with smaller representation from people who reported a household income under $15,000 (13.59%, n = 64) or over $200,000 (4.9%, n = 23). They were roughly equally split into dating/married (56.26%, n = 265) vs. being single, widowed, or divorced (43.74%, n = 206). Most respondents identified as heterosexual (88.32%, n = 416).

Table 2 shows descriptive statistics on all the variables of interest. For ease of interpretation we show the PROMIS scores, instead of the factor scores from the EFA, but it was the later we used in subsequent analyses. Participants showed relatively low levels of BPD traits; only 83 (17.62%) screened above the cut-off of clinically-significant borderline symptoms. Similarly, levels of poor work and social adjustment, depression, anger, and anxiety were relatively low, while well-being was relatively high, though there was variability in the endorsement of these measures. Most participants rated anxiety as the most difficult emotion to deal with (46.71%, n = 220), followed about equally by depression (21.86%, n = 103) and anger (20.59%, n = 97), with a minority of participants saying that all their emotions were out of control (10.83%, n = 51).

## WHO-5 well-being index

Overall, symptoms of depression, anxiety, and anger were associated with lower well-being, though borderline symptoms were not (see Table 3). Borderline symptoms interacted with anxiety, depression, and anger (see Table 4), but these effects were in the opposite direction of what we predicted. That is, symptoms of anxiety were associated with lower overall subjective well-being, but the effect became smaller as participants reported borderline symptoms increased. The Johnson-Neyman significance regions suggested that the reduction in the association between anxiety and low well-being transitioned to not statistically significant for individuals with ≥5.28 BPD traits. We computed simple slopes to illustrate the association between anxiety and well-being for individuals who reported no BPD traits and those who reported six traits (i.e., a plausible value higher than 5.28 on the MSI-BPD). As can be seen in Fig 1, for individuals who did not report BPD traits, increasing levels of anxiety were associated with decreased well-being. Individuals with six BPD traits reported overall lower levels of well-being but increasing symptoms of anxiety did not significantly predict decreases in well-being.

## Work and Social Adjustment Scale (WSAS)

Overall, symptoms of anxiety, depression, and anger predicted significantly more problems with work and social adjustment (see Table 3), though the effect sizes were generally small. There was an association between number of BPD traits and lower well-being, but this effect was small and not statistically significant. We found evidence for an interaction between BPD traits and anxiety that was in the direction we expected wherein anxiety was most strongly related to lower WSAS among participants high on BPD traits (see Table 4). Because this result contrasted with what we found on the WHO-5, and because the distribution of the WSAS was

**Table 2. Average levels of well-being, functioning, borderline, and internalizing symptoms.**

|  | Mean | SD | Me |
|---|---|---|---|
| Borderline Symptoms (MSI-BPD; 0–10) | 2.87 | 3.14 | 2 |
| Well-being (WHO-5; 0–100) | 56.54 | 24.17 | 58 |
| Work and Social Adjustment (WSAS; 0–40) | 10.07 | 11.74 | 5 |
| Depression (PROMIS; 0–100) | 56.40 | 10.95 | 57 |
| Anxiety (PROMIS; 0–100) | 56.99 | 10.52 | 58 |
| Anger(PROMIS; 0–100) | 53.21 | 12.36 | 53 |

Note. SD = standard deviation, Me = median, WHO-5 = WHO-5 Well-Being Index, WSAS = Work and Social Adjustment Scales, Depression = MSI-BPD = self-reported McLean Screening Instrument for Borderline Personality Disorder, PROMIS = Patient Reported Outcomes Measurement Information System—Emotional Distress

**Table 3. Multivariable prediction of well-being and work and social adjustment from symptoms of borderline personality disorder, depression, anger, and anxiety (n = 471).**

| Well-being (WHO-5) | B | SE | t | p value | β |
|---|---|---|---|---|---|
| Intercept | 57.00 | 1.10 | 51.97 | 0.000 | |
| BPD traits (MSI) | -0.80 | 0.45 | -1.79 | 0.075 | -0.10 |
| Depression (PROMIS) | -4.30 | 1.33 | -3.24 | 0.001 | -0.17 |
| Anger (PROMIS) | -3.04 | 1.27 | -2.40 | 0.017 | -0.11 |
| Anxiety (PROMIS) | -4.83 | 1.17 | -4.14 | 0.000 | -0.18 |
| Female (vs. male) | -8.18 | 2.06 | -3.97 | 0.000 | -0.34 |
| Work and social adjustment (WSAS) | B | SE | t | p value | β |
| Intercept | 9.30 | 0.46 | 20.22 | 0.000 | |
| BPD traits (MSI) | 0.78 | 0.19 | 4.13 | 0.000 | 0.21 |
| Depression (PROMIS) | 4.03 | 0.56 | 7.24 | 0.000 | 0.32 |
| Anger (PROMIS) | 2.53 | 0.53 | 4.75 | 0.000 | 0.20 |
| Anxiety (PROMIS) | 1.92 | 0.49 | 3.92 | 0.000 | 0.15 |
| Female (vs. male) | -3.12 | 0.86 | -3.61 | 0.000 | -0.27 |

Note. One woman, and two men, identified as transgender and were categorized according to their gender identity (i.e., female, and male respectively). WHO-5 = WHO-5 Well-Being Index, MSI-BPD = self-reported McLean Screening Instrument for Borderline Personality Disorder, Patient Reported Outcomes Measurement Information System, Emotional Distress

skewed, we conducted regression diagnostics to query the fit of the regression model. Visual inspection of the residuals from the model suggested that heteroskedasticity was present. When we conducted a similar linear regression using Huber-White standard errors, we could

**Table 4. Interactions between depression, anger, and anxiety and borderline traits in predicting well-being and work/social adjustment (n = 471).**

| Well-being (WHO-5) | B | SE | t | p value | β |
|---|---|---|---|---|---|
| Intercept | 53.80 | 1.23 | 43.59 | 0.000 | |
| BPD traits (MSI) | -1.50 | 0.46 | -3.26 | 0.001 | -0.20 |
| Depression (PROMIS) | -5.86 | 1.36 | -4.32 | 0.000 | -0.19 |
| Anger (PROMIS) | -4.84 | 1.34 | -3.61 | 0.000 | -0.14 |
| Anxiety (PROMIS) | -5.80 | 1.16 | -5.00 | 0.000 | -0.19 |
| Female (vs. male) | -7.72 | 2.01 | -3.85 | 0.000 | -0.32 |
| BPD * depression | 1.12 | 0.39 | 2.86 | 0.004 | 0.14 |
| BPD * anger | 1.16 | 0.36 | 3.23 | 0.001 | 0.14 |
| BPD * anxiety | 0.84 | 0.38 | 2.24 | 0.026 | 0.10 |
| Work and social adjustment (WSAS) | B | SE | t | p value | β |
| Intercept | 9.05 | 0.53 | 17.08 | 0.000 | |
| BPD traits (MSI) | 0.73 | 0.20 | 3.70 | 0.000 | 0.20 |
| Depression (PROMIS) | 4.07 | 0.58 | 6.98 | 0.000 | 0.32 |
| Anger (PROMIS) | 2.19 | 0.58 | 3.80 | 0.000 | 0.18 |
| Anxiety (PROMIS) | 1.72 | 0.50 | 3.45 | 0.001 | 0.16 |
| Female (vs. male) | -3.06 | 0.86 | -3.55 | 0.000 | -0.26 |
| BPD * depression | -0.13 | 0.17 | -0.77 | 0.442 | -0.03 |
| BPD * anger | 0.20 | 0.15 | 1.33 | 0.185 | 0.05 |
| BPD * anxiety | 0.33 | 0.16 | 2.01 | 0.045 | 0.08 |

Note. WHO-5 = WHO-5 Well-Being Index, MSI-BPD = self-reported McLean Screening Instrument for Borderline Personality Disorder, Patient Reported Outcomes Measurement Information System, Emotional Distress

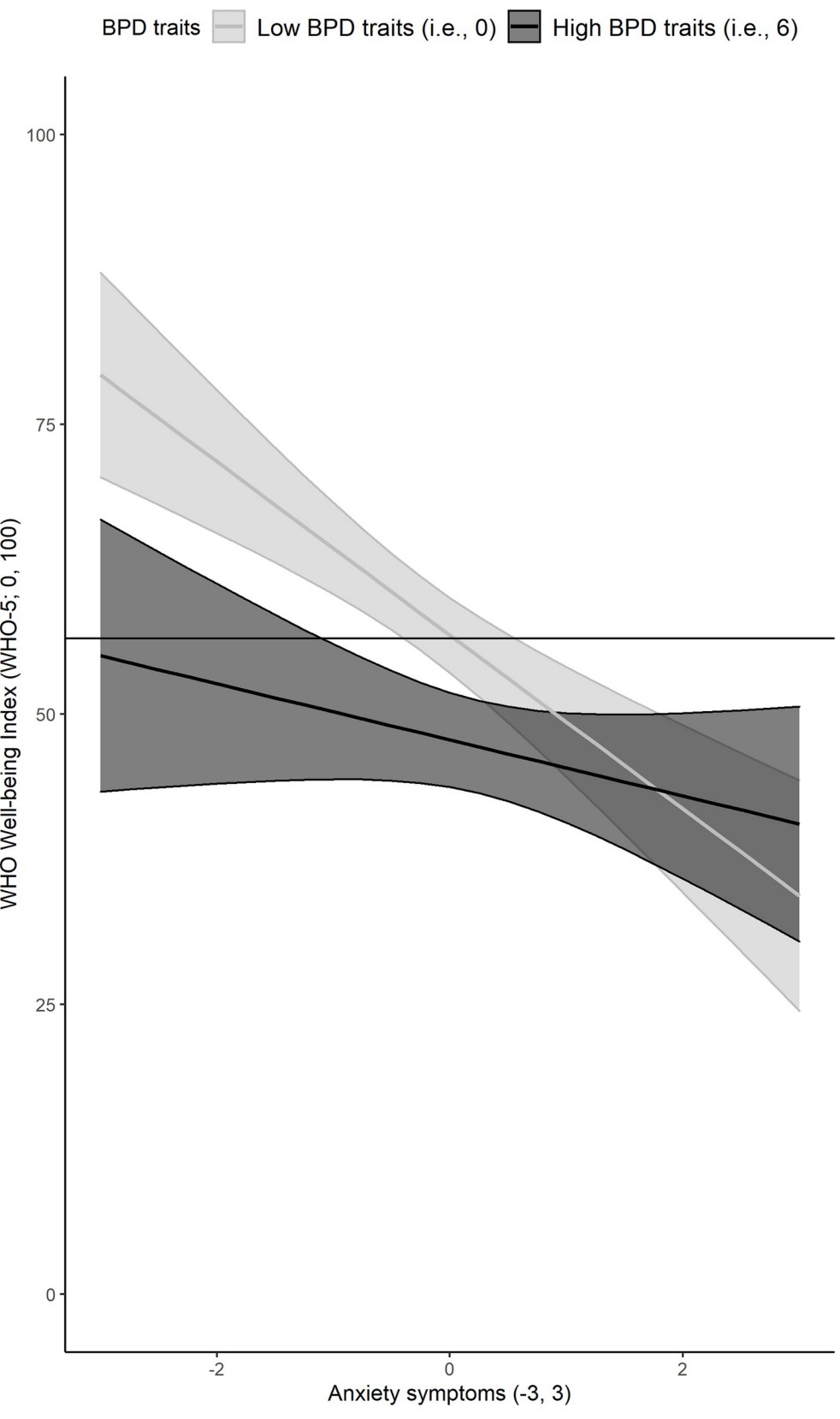

**Fig 1. Predicted association between anxiety symptoms and well-being for patients high (i.e., 6) vs. low (i.e., 0) on Borderline Personality Disorder (BPD) traits (N = 471).** Note: Confidence bands represent 95% confidence intervals on the prediction estimates with continuos variable (i.e., depression and anger) set at their mean and gender as the average between women and men. Horizontal black line indicates the sample mean.

not find evidence that anxiety interacted with BPD traits to predict WSAS (B = 0.33, SE = 0.20, t(462) = 1.63, $p$ = 0.10).

## Self-reported difficulty with emotions

We captured responses about which emotions were most difficult to deal with: depression, anxiety, anger, or all negative emotions (i.e., all my emotions are out of control). As previously stated, anxiety was the emotion most participants struggled with (46.71%) with a minority of individuals reporting that they have difficulty with all emotions. When we calculated the mean number of BPD traits by which emotions participants reported struggling with the most, we found that the self-report of which emotions is most difficult predicts BPD traits (F(469, 1) = 15.07, p < 0.0001).

Specifically, reporting difficulty controlling "all emotions" was associated with a greater number of BPD traits (M = 5.82, SD = 3.37), compared to reporting struggling more with anxiety (M = 2.12, SD = 2.88) depression (M = 3.25, SD = 3.06) or anger (M = 2.60, SD = 2.72). To rule out the possibility that it was only the participants who reported an answer other than "all my emotions are out of control" who may have specific difficulties with anxiety, we re-ran this analysis, excluding participants who said "all my emotions are out of control." When doing this, the number of BPD traits an individual endorsed was not associated with the single emotion they reported struggling with the most (F(418, 1) = 2.72, p = 0.10).

## Discussion

BPD traits, which generally reflect difficulty managing emotions along with broad personality disturbances, and symptoms of depression and anxiety form part of the internalizing spectrum of psychopathology [46, 47]. Studies suggest high levels of comorbidity between BPD and mood and anxiety symptoms [20], with some data to suggest that anxiety comorbidity *per se* may signal vulnerability to BPD relative to other personality disturbances [20, 21, 25, 26]. Nonetheless, there has also been a paucity of research assessing anxiety symptoms in individuals with BPD symptomatology. The current study sought to contribute to this literature by examining the interactions between symptoms of BPD and anxiety in predicting well-being and functioning.

Contrary to our hypothesis, our data do not support the idea that anxiety contributes to poorer well-being and worse functioning as BPD traits increase. We found no support for this idea when participants were queried directly nor when exploring the interaction of BPD traits and anxiety in predicting well-being and functioning. In fact, the interaction between anxiety and BPD traits in predicting well-being on the WHO-5 was in the opposite direction: anxiety was less predictive of well-being at higher levels of BPD traits.

Several limitations should be considered before interpreting the results of this study. The first of which is that data collection occurred in the beginning of a worldwide pandemic, in the dates 4/16/20-4/20/20, which may have skewed the results. It is unknown the degree to which the pandemic may impact features of the sample, both in terms of types of respondents and also in terms of symptom severity and their effect on functionality and well-being. Although the context of the pandemic may have influenced overall levels of severity represented in the sample, it is unclear whether it would have biased our specific question: whether anxiety and

BPD traits interacted to predict well-being and functioning. Despite our best efforts to create a representative sample, our sample is not very representative in terms of education and race/ethnicity, which limits generalizability of results. Additionally, the mean age was 47, which is older than average age of peak BPD symptoms and may have impacted our results.

Second, although we attempted to exclude bots, it is possible that our data are nonetheless not of optimal quality. Moreover, it is possible that our sample did not include enough individuals high on BPD traits. We only measured BPD traits via self-report which may not be optimal and ignores any biological vulnerability to psychopathology. Finally, our data are cross-sectional, which does not capture the temporal dynamics of BPD traits [48], anxiety, and functioning.

Despite limitations to our study, we collected data on a variety of measures to probe whether there was an interaction between BPD traits and anxiety in predicting outcome. We measured two different aspects of quality of life: subjective-well-being and work and social functioning. Moreover, we undertook this analysis while controlling for other symptoms of negative affectivity, namely anger and depression. Overall, BPD symptoms were associated with lower well-being and more difficulties with work and social adjustment. The higher BPD traits reported by individuals, the less anxiety was associated to poor well-being, though the same pattern of results was not present in relation to work and social adjustment. It is possible that our results reflect a ceiling effect of the impacts of anxiety on well-being among people with higher levels of BPD traits. Because our data are cross-sectional, another interpretation equally supported by the data is that at high levels of anxiety, BPD traits were not predictive of lower well-being. A parsimonious explanation of these findings may be that impairment in functioning in BPD and comorbid anxiety is less a result of the additive impact of having anxiety on top of BPD traits, and more a result of the severity of more distal risk factors like overall negative emotional emotionality. Indeed, structural studies of psychopathology have indicated that core dimensions such as high neuroticism and poor emotion regulation abilities result in numerous mental health outcomes including BPD, anxiety, and their co-morbidity [49–53]. Future research should explore the temporal relationships between BPD symptoms and specific effects of negative emotionality on functioning. Despite potentially different interpretations of our findings, it is worth noting that anxiety was the emotion most participants reported they struggled with. This is consistent with the primal role that anxiety has in prominent theories of psychopathology like psychoanalytic theory as well as the theories that underlie acceptance and commitment therapy [54]. It is, then, troublesome, that a BPD diagnosis excludes individuals from some anxiety treatment trials. Our results suggest that individuals higher on BPD traits are like other individuals in that anxiety is the most commonly-reported emotion they have difficulty regulating.

In our analyses of the WHO-5, we found that the effects of anxiety on well-being decreased as the level of BPD traits increased. One possible explanation for these findings is that as BPD traits increase, it is the BPD traits in and of themselves that are more predictive of lower well-being symptoms of negative affectivity. Conversely, we found no such pattern on the WSAS. The WSAS and the WHO-5 were selected because they both capture aspects of mental health that go above and beyond a focus on illness, however, the measures were poorly correlated in this sample. While the WHO-5 captures positive affectivity (e.g., "I have felt cheerful and in good spirits"), the WSAS is targeted towards specific ways in which a problem, in our case emotion dysregulation, affects areas of life like friendships and work.

Our hypothesis that individuals with BPD features are more impaired by symptoms of anxiety than those without BPD features was not supported, at least as captured by the cross-sectional data on the measures we administered. Nonetheless, we did find some evidence that the effect of anxiety on measures of functioning may vary across BPD traits. Additionally, the

context that triggers the emotional response may be another factor to consider [48]. Emotion regulation is a dynamic and complex process. Future research should explore the temporal dynamics of different emotions as well as their relative contributions to psychosocial functioning and well-being.

## Supporting information

**S1 Table. Exploratory Factor Analysis (EFA) of PROMIS depression, anxiety, and anger (N = 471).** Note. PROMIS = Patient Reported Outcomes Measurement Information System. Factor loading from an exploratory factor analysis with ordinary least squares estimation and three-factor varimax rotation.
(PDF)

**S2 Table. Interactions between Borderline Personality Disorder (BPD) traits and depression, anger, and anxiety and in predicting well-being and work/social adjustment while adding interactions of gender with depression, anger, anxiety, and BPD traits (n = 471).**
(PDF)

**S3 Table. Interactions between depression, anger, and anxiety and borderline traits in predicting well-being and work/social adjustment (n = 471), including Variance Inflation Factors (VIFs).** Note. VIF = variance inflation factor, WHO-5 = WHO-5 Well-Being Index, MSI-BPD = self-reported McLean Screening Instrument for Borderline Personality Disorder, Patient Reported Outcomes Measurement Information System, Emotional Distress.
(PDF)

## Author Contributions

**Conceptualization:** Jacqueline Howard, Allison Peipert, Lorenzo Lorenzo-Luaces.

**Data curation:** Robinson De Jesu´s-Romero, Lorenzo Lorenzo-Luaces.

**Formal analysis:** Robinson De Jesu´s-Romero, Lorenzo Lorenzo-Luaces.

**Investigation:** Jacqueline Howard, Robinson De Jesu´s-Romero, Lorenzo Lorenzo-Luaces.

**Methodology:** Jacqueline Howard, Lorenzo Lorenzo-Luaces.

**Project administration:** Jacqueline Howard, Lorenzo Lorenzo-Luaces.

**Resources:** Lorenzo Lorenzo-Luaces.

**Software:** Lorenzo Lorenzo-Luaces.

**Supervision:** Lorenzo Lorenzo-Luaces.

**Visualization:** Lorenzo Lorenzo-Luaces.

**Writing – original draft:** Jacqueline Howard, Allison Peipert, Tennisha Riley, Lauren A. Rutter, Lorenzo Lorenzo-Luaces.

**Writing – review & editing:** Jacqueline Howard, Robinson De Jesu´s-Romero, Allison Peipert, Tennisha Riley, Lauren A. Rutter, Lorenzo Lorenzo-Luaces.

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
