## [Decision Letter · Decision Letter 0]

24 Sep 2020

PONE-D-20-25421

The significance of anxiety symptoms in predicting psychosocial functioning across borderline personality traits

PLOS ONE

Dear Dr. Luaces,

Thank you for submitting your manuscript to PLOS ONE. After careful consideration, we feel that it has merit but does not fully meet PLOS ONE’s publication criteria as it currently stands. Therefore, we invite you to submit a revised version of the manuscript that addresses the points raised during the review process.

We look forward to receiving your revised manuscript.

Kind regards,

Stephan Doering, M.D.

Academic Editor

PLOS ONE

2. Please provide additional details regarding participant consent.

In the Methods section, please ensure that you have specified (i) whether consent was informed and (ii) what type you obtained (for instance, written or verbal).

If your study included minors, state whether you obtained consent from parents or guardians.

If the need for consent was waived by the ethics committee, please include this information.

Reviewers' comments:

Reviewer's Responses to Questions

**Comments to the Author**

1. Is the manuscript technically sound, and do the data support the conclusions?

Reviewer #1: Yes

Reviewer #2: No

Reviewer #3: Yes

2. Has the statistical analysis been performed appropriately and rigorously? 

Reviewer #1: Yes

Reviewer #2: No

Reviewer #3: Yes

3. Have the authors made all data underlying the findings in their manuscript fully available?

Reviewer #1: Yes

Reviewer #2: Yes

Reviewer #3: Yes

4. Is the manuscript presented in an intelligible fashion and written in standard English?

Reviewer #1: Yes

Reviewer #2: No

Reviewer #3: Yes

5. Review Comments to the Author

Reviewer #1: This brief manuscript examines relationships amongst BPD symptoms, anxiety, depression, and anger and well-being/functioning. Given that some have argued that BPD is a disorder of emotion dysregulation, it is valuable to understand how discrete emotional experiences contribute to this clinical presentation. Although I think this manuscript makes an interesting contribution, I had a number of conceptual questions that may underscore why the authors did not find their hypothesized relationships amongst these variables. 

First, the premise that anxiety is relevant for disorders that are maintained by emotion regulation deficits (e.g., BPD) makes sense since anxiety is an emotion. The authors argue the anxiety is not specifically captured in the BPD diagnostic criteria, however anxiety would be captured in mood lability. Because anxiety is so common (as the authors note), it doesn't really differentiate BPD from other emotional disorders, whereas anger might. Thus, lack of specific attention to anxiety in the BPD diagnostic criteria is an artifact of a categorical classification (including criteria that differentiates disorders), but is not necessarily as problematic as the authors suggest. 

Additionally, since BPD and anxiety disorders develop from similar functional mechanisms, it makes sense that symptoms would co-occur. However, it seems the authors are thinking about comorbidity amongst BPD and anxiety disorders as having an additive effect on functioning. However, functioning is probably more related to severity of the risk factor (e.g., neuroticism, emotional vulnerability) rather than the discrete symptoms (anxiety, avoidance, interpersonal difficulties, impulsive behaviors) that manifest phenotypically. It would be useful to consider these conceptual issues in the manuscript's introduction and discussion.

In addition, there were a few minor points to address:

Why is the general description of the PROMIS included under the BPD symptoms subheading?

In the planned analysis, I'm not sure what is meant by "the results of these analyses reveal the overall effects of internalizing/externalizing on functioning." Are anger and BPD features meant to represent externalizing? I'm not sure this is supported by your introduction? Anger is an emotion - so it could very well be situated on internalizing. BPD represents a confluence of internalizing and externalizing, though some have found that for some patients, BPD symptoms can be expressed entirely via internalizing (Eaton et al., 2011).

I am pretty sure that the incorrect range for possible ZAN-BPD scores is reported (on this measure, each BPD symptom is rated on 0-4 Likert Scale).

Reviewer #2: The authors examined the role of anxiety symptoms in the context of BPD in predicting psychosocial functioning in a non-clinical population of adults.

I agree with the authors that the role of anxiety in BPD needs to be examined. Nevertheless, there are several problems with the study.

1. The analytic approach is difficult to follow and am I not sure if it’s the right way to go. I expected to see a traditional moderating analyses because the literature review was leading in that direction. I think a simpler approach is to look at anxiety as the moderator between BPD and well-being.

2. The moderating role of sex/gender needs to be examined given the higher likelihood of BPD and anxiety in women compared to men.

3. The average age of sample is close to 47. I wonder what the implications are of this. BPD symptoms (severity and type) tend to vary with age. Some discussion is needed.

4. The discussion falls short. The discrepant findings need to be explained and not just re-stated.

5. The introduction is a little all over the place. Relatedly, the writing needs some improvement. The authors change verb tense throughout, as one example.

Reviewer #3: The authors examine the independent and interactive effects of BPD traits and anxiety in predicting well-being and work and social adjustment in a sample of 471 online respondents. They find that both BPD and anxiety contributed to poorer well-being, but they also interacted such that anxiety was only predictive of wellbeing at relatively low levels of BPD traits. Strengths of the manuscript include careful attention to many methodological details, including sampling concerns and missing data. As the authors note, the manuscript fills a need in examining how anxiety contextualizes the impact of anxiety, and vice versa. I did have a few questions for clarification and some suggestions for improvement.

Minor concern: The sample is not very representative in terms of education and race/ethnicity.

Were the EFA factors used in regression analyses? I could not tell, although table 4 seems to suggest that PROMIS scale scores were used. This is important, I think, to clarify because it is unclear how the authors dealt with dependency among depression, anger, and anxiety (which is considerable). Because these variables also relate to wellbeing, “controlling for” them might produce paradoxical results (cf. Lord’s, or Simpson’s, paradoxes). Varimax rotation (which I think produces orthogonal factors) might correct for this, but then the factors are exploratory, so the overlap is perhaps still present, just pushed down to the item level. Some more clarity would be helpful.

Similarly, to what degree is collinearity between anxiety and BPD a concern? The authors report a moderately strong correlation between anxiety and BPD, but I could not figure out if they had accounted for this dependency prior to their regression analyses. In general, they might consider reporting VIFs or dealing with multicollinearity in some other way (e.g., by centering predictor variables), or both.

Although the results ran contrary to hypotheses, with anxiety conveying little information about wellbeing at high levels of borderline pathology, it would be helpful if the authors could unpack these results a bit more. Why is this occurring? They mention a possible “ceiling effect” (p. 9), whereby the effect of additional anxiety is crowded out by borderline pathology. I think this is plausible. They also speculate that “as BPD traits increase, it is the BPD traits in and of themselves that are more predictive of lower well-being…”. I am not sure how different this is from the ceiling effect idea. But is the interaction effect not equally interpretable the other way, where BPD traits are only predictive of well-being at low levels of anxiety? If so, that might argue in favor of the ceiling effect idea, but not of the idea that BPD traits have primacy over anxiety in predicting wellbeing as they increase. Perhaps the ceiling effect simply pertains to overall, general psychopathology – at low levels, increases signal increasing impairment, but this increase is negatively accelerated as psychopathology increases?

6. PLOS authors have the option to publish the peer review history of their article (what does this mean?). If published, this will include your full peer review and any attached files.

Reviewer #1: No

Reviewer #2: No

Reviewer #3: **Yes: **William D. Ellison

---

## [Author Response · Author response to Decision Letter 0]

16 Nov 2020

Response to reviews

Editor:

Response: We have renamed the files to be in accord to the journal requirements. 

2. Please provide additional details regarding participant consent.

In the Methods section, please ensure that you have specified (i) whether consent was informed and (ii) what type you obtained (for instance, written or verbal).

If your study included minors, state whether you obtained consent from parents or guardians.

If the need for consent was waived by the ethics committee, please include this information.

Response: We have added language to the methods section clarifying the informed consent.

Response: We have added a caption for the figure.

Response: We have reformatted the supporting information in accordance with the journal’s style.

Reviewer's Responses to Questions

Comments to the Author

1. Is the manuscript technically sound, and do the data support the conclusions?

Reviewer #1: Yes

Reviewer #2: No

Reviewer #3: Yes

2. Has the statistical analysis been performed appropriately and rigorously? 

Reviewer #1: Yes

Reviewer #2: No

Reviewer #3: Yes

3. Have the authors made all data underlying the findings in their manuscript fully available?

Reviewer #1: Yes

Reviewer #2: Yes

Reviewer #3: Yes

4. Is the manuscript presented in an intelligible fashion and written in standard English?

Reviewer #1: Yes

Reviewer #2: No

Reviewer #3: Yes

5. Review Comments to the Author

Reviewer #1: This brief manuscript examines relationships amongst BPD symptoms, anxiety, depression, and anger and well-being/functioning. Given that some have argued that BPD is a disorder of emotion dysregulation, it is valuable to understand how discrete emotional experiences contribute to this clinical presentation. Although I think this manuscript makes an interesting contribution, I had a number of conceptual questions that may underscore why the authors did not find their hypothesized relationships amongst these variables. 

Response: We thank the reviewer for noting that our manuscript makes an interesting contribution to the literature.

First, the premise that anxiety is relevant for disorders that are maintained by emotion regulation deficits (e.g., BPD) makes sense since anxiety is an emotion. The authors argue the anxiety is not specifically captured in the BPD diagnostic criteria, however anxiety would be captured in mood lability. Because anxiety is so common (as the authors note), it doesn't really differentiate BPD from other emotional disorders, whereas anger might. Thus, lack of specific attention to anxiety in the BPD diagnostic criteria is an artifact of a categorical classification (including criteria that differentiates disorders), but is not necessarily as problematic as the authors suggest. 

Response: The reviewer brings up an interesting point that anxiety is, to some extent, captured in the BPD diagnosis because of symptoms related to mood lability. Our original point was that the BPD criteria specifically mention anger and depressed mood (e.g., emptiness, suicidal behaviors) above and beyond their inclusion as part of mood liability. The data on how anxiety comorbidity differentiates BPD from other PDs also suggests that there is some degree of importance to anxiety comorbidity in BPD. An additional point to consider is that high anxiety and mood lability could independent. For example, a person with high trait anxiety may have persistent feelings of emptiness and irritability, but low mood lability and nonetheless meet criteria for BPD. Nonetheless, we agree more than disagree with the reviewer that the lack of specification may be an artifact of categorical classification, and have added this to the manuscript, pp. 2.

“Anxiety not being a part of the diagnostic criteria of BPD may simply be an artifact of DSM-5 categorical classification. However, it is curious…”

Additionally, since BPD and anxiety disorders develop from similar functional mechanisms, it makes sense that symptoms would co-occur. However, it seems the authors are thinking about comorbidity amongst BPD and anxiety disorders as having an additive effect on functioning. However, functioning is probably more related to severity of the risk factor (e.g., neuroticism, emotional vulnerability) rather than the discrete symptoms (anxiety, avoidance, interpersonal difficulties, impulsive behaviors) that manifest phenotypically. It would be useful to consider these conceptual issues in the manuscript's introduction and discussion.

We thank the reviewer for bringing up this point. We have added these considerations to the discussion: 

“A parsimonious explanation of our findings may be that impairment in functioning in BPD and comorbid anxiety is less a result of the additive impact of having anxiety on top of BPD traits, and more a result of the severity of more distal risk factors like overall negative emotional emotionality. Indeed, structural studies of psychopathology have indicated that core dimensions such as high neuroticism and poor emotion regulation abilities result in numerous mental health outcomes including BPD, anxiety, and their co-morbidity (e.g., Brandes & Tackett, 2019; Eaton et al., 2011; Eftekhari et al., 2009; Gross & John, 2003; Ormel et al., 2013).”

In addition, there were a few minor points to address:

Why is the general description of the PROMIS included under the BPD symptoms subheading?

Response: This was a formatting error. We have added specific subheading to refer to symptoms of depression, anxiety, and anger. We thank the reviewer for catching this error.

In the planned analysis, I'm not sure what is meant by "the results of these analyses reveal the overall effects of internalizing/externalizing on functioning." Are anger and BPD features meant to represent externalizing? I'm not sure this is supported by your introduction? Anger is an emotion - so it could very well be situated on internalizing. BPD represents a confluence of internalizing and externalizing, though some have found that for some patients, BPD symptoms can be expressed entirely via internalizing (Eaton et al., 2011).

Response: We removed the sentence in question, we agree it was confusing and unnecessary.

I am pretty sure that the incorrect range for possible ZAN-BPD scores is reported (on this measure, each BPD symptom is rated on 0-4 Likert Scale).

Response: This was an error in our writing. As the reviewer notes, the BPD screener by Zanarini which is usually referred to as the “Zanarini BPD Scale” provides a continuous measure with items rated on a 0-4 Likert scale is available both as an interviewer-rated measure and as a self-reported measure. This is not the measure we used. We used the self-reported BPD screener that was also developed by Zanarini, which is a 10-item true/false self-reported instrument. (The confusion on our end stemmed from one of us referencing the “Zanarini scale” despite the author formally naming the scale “The McLean Screening Instrument for BPD (MSI-BPD).” We made the changes to the text throughout and thank the review for catching this error.)

Reviewer #2: The authors examined the role of anxiety symptoms in the context of BPD in predicting psychosocial functioning in a non-clinical population of adults.

I agree with the authors that the role of anxiety in BPD needs to be examined. Nevertheless, there are several problems with the study.

1. The analytic approach is difficult to follow and am I not sure if it’s the right way to go. I expected to see a traditional moderating analyses because the literature review was leading in that direction. I think a simpler approach is to look at anxiety as the moderator between BPD and well-being.

Response: We followed a “traditional” moderator analysis in which we explored the interaction between anxiety and BPD in predicting the functioning measures. A part of our analysis that we situated in the “Results” section was using factor scores from an EFA to isolate variance unique to anxiety, depression, or anger. There was some text from an old version of the manuscript which we removed and may have been the source of confusion. In terms of whether to conceptualize anxiety as the moderator or BPD as the moderator, the cross-sectional nature of our study makes it hard to establish any kind of causal precedence over one or the other. We have revised the methods, including to reference the test of moderation but are happy to make other specific changes.

2. The moderating role of sex/gender needs to be examined given the higher likelihood of BPD and anxiety in women compared to men.

Response: Our main research question did not involve exploring the role of gender/sex. As the reviewer notes, BPD traits and depression and anxiety tend to be higher, on average, in women compared to men. We bring to the reviewer’s attention that our research question concerns an interaction (i.e., among individuals with high BPD traits anxiety would predict poorer functioning than among individuals with low BPD traits) not a focus on average levels. With this context, we are aware of no research suggesting that gender/sex moderates the association of BPD with functioning or anxiety with functioning or even BPD on anxiety. Nonetheless, to address the reviewer’s point, we made two changes to our analytic plan. First, in all analyses that are presented in the main text, we control for gender which we code in the binary women (+0.5) vs. men (-0.5). (Three individuals identified as transgender, one trans woman and two trans men and were coded as women and men, respectively). Next, to address the reviewers concern, we include in the supplementary analysis interactions between gender and anxiety and gender and gender borderline symptoms. (The reviewer did not specify what they meant by “moderating role of sex/gender” which left it open for interpretation.) Including terms for gender interacting with BPD, anxiety, depression, and anger did not change the pattern of results. The reviewer may be interested to know that sex and BPD traits interacted to predict subjective well-being, but not work and social adjustment. However, because this was not part of our research question, and because they did not change the results of the main variables we cared about, we do not wish to report these results in the main text. One of our concerns is that readers may get the impression that we conducted various exploratory tests (e.g., “p-hacking”).

3. The average age of sample is close to 47. I wonder what the implications are of this. BPD symptoms (severity and type) tend to vary with age. Some discussion is needed.

Response: To our knowledge, the developmental trend for BPD symptoms is a large reduction in behavioral dysregulation and relative stability in interpersonal dysregulation with intermediate changes with emotional dysregulation. Thus, it is not entirely obvious how the developmental state of our sample may affect our results. Nonetheless, to address the reviewer’s desire for a discussion of this topic, we included a sentence in the discussion: “Additionally, the mean age was 47, which is older than average age of peak BPD symptoms and may have impacted our results.” 

4. The discussion falls short. The discrepant findings need to be explained and not just re-stated.

Response: We have expanded our discussion in several areas, noting the importance of individual differences including age, education, race/ethnicity, and adding more studies about the structure of psychopathology as it relates to BPD features. As well, we bring up alternative explanations for our findings as suggested by other reviewers. We thank the reviewer for bringing up these concerns. One of the things that makes it a bit difficult to expand on the discussion is that we did not find support for our hypothesis but only found partial support for the opposite conclusion. Thus, we don’t want to express a false sense of certainty over the pattern of the findings.

5. The introduction is a little all over the place. Relatedly, the writing needs some improvement. The authors change verb tense throughout, as one example.

Response: We have generally edited the introduction for readability and closely edited it by paying attention to verb tense.

Reviewer #3: The authors examine the independent and interactive effects of BPD traits and anxiety in predicting well-being and work and social adjustment in a sample of 471 online respondents. They find that both BPD and anxiety contributed to poorer well-being, but they also interacted such that anxiety was only predictive of wellbeing at relatively low levels of BPD traits. Strengths of the manuscript include careful attention to many methodological details, including sampling concerns and missing data. As the authors note, the manuscript fills a need in examining how anxiety contextualizes the impact of anxiety, and vice versa. I did have a few questions for clarification and some suggestions for improvement.

Minor concern: The sample is not very representative in terms of education and race/ethnicity.

Response: We thank the reviewer for bringing this up, we now address it in our discussion as a limitation.

Were the EFA factors used in regression analyses? I could not tell, although table 4 seems to suggest that PROMIS scale scores were used. This is important, I think, to clarify because it is unclear how the authors dealt with dependency among depression, anger, and anxiety (which is considerable). Because these variables also relate to wellbeing, “controlling for” them might produce paradoxical results (cf. Lord’s, or Simpson’s, paradoxes). Varimax rotation (which I think produces orthogonal factors) might correct for this, but then the factors are exploratory, so the overlap is perhaps still present, just pushed down to the item level. Some more clarity would be helpful.

Response: Yes, the EFA factors were used in the regression analysis. On table 4 we presented the raw scores to give the reader a clear understanding of the absolute levels of depression, anxiety, and anger that were represented in the sample. (The EFA scores will be centered on 0 so that’s less helpful to present.) We have clarified this in the text

Similarly, to what degree is collinearity between anxiety and BPD a concern? The authors report a moderately strong correlation between anxiety and BPD, but I could not figure out if they had accounted for this dependency prior to their regression analyses. In general, they might consider reporting VIFs or dealing with multicollinearity in some other way (e.g., by centering predictor variables), or both.

Response: The anxiety scores are already centered on 0 because they are factors scores. (This was noted in the text.) As per the reviewers suggestion, we centered the BPD screening scale on its median score (as per Kraemer and Blasey’s 2004 paper). Similarly, gender is also centered (in the +/- 0.5 system, a binary variable being centered on zero means that the outcome is interpreted as being at the average between women and men). 

Additionally, we report the VIFs in a supplement. They ranged from 1.09 to 2.12. Thus suggesting that collinearity is not a major issue in these analyses. The reviewer, however, is correct that anxiety and BPD features correlate. The factor scores are correlated from 0.27 (anxiety) to 0.54 (depression) We mention in the discussion that future research should employ longitudinal and experimental methods to disentangle unique effects of anxiety above and beyond borderline personality symptoms.

Although the results ran contrary to hypotheses, with anxiety conveying little information about wellbeing at high levels of borderline pathology, it would be helpful if the authors could unpack these results a bit more. Why is this occurring? They mention a possible “ceiling effect” (p. 9), whereby the effect of additional anxiety is crowded out by borderline pathology. I think this is plausible. They also speculate that “as BPD traits increase, it is the BPD traits in and of themselves that are more predictive of lower well-being…”. I am not sure how different this is from the ceiling effect idea. But is the interaction effect not equally interpretable the other way, where BPD traits are only predictive of well-being at low levels of anxiety? If so, that might argue in favor of the ceiling effect idea, but not of the idea that BPD traits have primacy over anxiety in predicting wellbeing as they increase. Perhaps the ceiling effect simply pertains to overall, general psychopathology – at low levels, increases signal increasing impairment, but this increase is negatively accelerated as psychopathology increases?

Response: The reviewer raises a crucial point regarding the direction of causality and the interpretation of our results. We acknowledge the limitation in our cross-sectional analysis which cannot disentangle causal relations between variables but generally is not supportive of our original hypothesis. It is worth nothing that the ratings of anxiety were collected in reference to a one-week period whereas the BPD screener queries “the past several years.” Thus, we expect that it is the more proximal state anxiety that modifies the association between anxious state and perceived well-being. However, given that these results run counter to our hypothesis and were not supported by the analyses with the WSAS, we do not wish to elaborate extensively on them. We have, however, added a sentence to raise the interpretation favored by the reviewer. 

“Because our data are cross-sectional, another interpretation equally supported by the data is that at high levels of anxiety, BPD traits were not predictive of lower well-being.”

And

“Future research should explore the temporal relationships between BPD symptoms and specific effects of negative emotionality on functioning.”

6. PLOS authors have the option to publish the peer review history of their article (what does this mean?). If published, this will include your full peer review and any attached files.

Do you want your identity to be public for this peer review? For information about this choice, including consent withdrawal, please see our Privacy Policy.

Reviewer #1: No

Reviewer #2: No

Reviewer #3: Yes: William D. Ellison

---

## [Decision Letter · Decision Letter 1]

22 Dec 2020

The significance of anxiety symptoms in predicting psychosocial functioning across borderline personality traits

PONE-D-20-25421R1

Dear Dr. Lorenzo Luaces,

We’re pleased to inform you that your manuscript has been judged scientifically suitable for publication and will be formally accepted for publication once it meets all outstanding technical requirements.

Kind regards,

Stephan Doering, M.D.

Academic Editor

PLOS ONE

Reviewers' comments:

Reviewer's Responses to Questions

**Comments to the Author**

1. If the authors have adequately addressed your comments raised in a previous round of review and you feel that this manuscript is now acceptable for publication, you may indicate that here to bypass the “Comments to the Author” section, enter your conflict of interest statement in the “Confidential to Editor” section, and submit your "Accept" recommendation.

Reviewer #1: All comments have been addressed

Reviewer #3: All comments have been addressed

2. Is the manuscript technically sound, and do the data support the conclusions?

Reviewer #1: Yes

Reviewer #3: Yes

3. Has the statistical analysis been performed appropriately and rigorously? 

Reviewer #1: Yes

Reviewer #3: Yes

4. Have the authors made all data underlying the findings in their manuscript fully available?

Reviewer #1: Yes

Reviewer #3: Yes

5. Is the manuscript presented in an intelligible fashion and written in standard English?

Reviewer #1: Yes

Reviewer #3: Yes

6. Review Comments to the Author

Reviewer #1: The authors have addressed my concerns and I think this manuscript will make an interesting contribution to the literature

Reviewer #3: (No Response)

7. PLOS authors have the option to publish the peer review history of their article (what does this mean?). If published, this will include your full peer review and any attached files.

Reviewer #1: **Yes: **Shannon Sauer-Zavala

Reviewer #3: **Yes: **William D Ellison

---

## [Editor Report · Acceptance letter]

4 Jan 2021

PONE-D-20-25421R1 

The significance of anxiety symptoms in predicting psychosocial functioning across borderline personality traits 

Dear Dr. Lorenzo-Luaces:

I'm pleased to inform you that your manuscript has been deemed suitable for publication in PLOS ONE. Congratulations! Your manuscript is now with our production department. 

Kind regards, 

on behalf of

Professor Stephan Doering 

Academic Editor

PLOS ONE